# Analysis of Genetic Variants MTHFR C677T, ACE I/D, AT1R A1166C and eNOS 4a/b in the Context of Essential Hypertension Susceptibility

**DOI:** 10.3390/biomedicines13112807

**Published:** 2025-11-18

**Authors:** Remus Nica, Silvia Nica, Luciana Teodora Rotaru, Mihai Toma, Lavinia Mariana Berca, Dănuț Cimponeriu, Roxana Măciucă

**Affiliations:** 1Faculty of Midwifery and Nursing, “Carol Davila” University of Medicine and Pharmacy, 020021 Bucharest, Romania; remus.nica@umfcd.ro; 2“Dr. Carol Davila” Central Military Emergency University Hospital, 010825 Bucharest, Romania; mihai.toma@prof.utm.ro; 3Faculty of Medicine, “Carol Davila” University of Medicine and Pharmacy, 020021 Bucharest, Romania; silvia.nica@umfcd.ro; 4University Emergency Hospital of Bucharest, 050098 Bucharest, Romania; 5Faculty of Medicine, University of Medicine and Pharmacyof Craiova, 200349 Craiova, Romania; lucianarotaru@yahoo.com; 6Molecular Biology Laboratory, National Institute of Research and Development for Food Bioresources, 021102 Bucharest, Romania; laviniamariana.berca@bioresurse.ro; 7Faculty of Biology, University of Bucharest, 050095 Bucharest, Romania; maciuca.roxana-alexandra22@s.bio.unibuc.ro

**Keywords:** essential hypertension, single nucleotide polymorphism, MTHFR C677T, AT1R A1166C, ACE I/D, eNOS 4a/b

## Abstract

Arterial hypertension (AH) is an important risk factor for cardiovascular diseases, a group of diseases that constitutes the most frequent cause of death worldwide. Most AH patients globally are diagnosed with essential hypertension (EH), since they do not present an identifiable cause for high blood pressure (HBP). The aim of this study was to assess the associations between EH and genetic variants MTHFR C677T, ACE I/D, AT1R A1166C and eNOS 4a/b in the adult Caucasian population of Romania. **Methods:** A case–control study was conducted including 845 EH patients and 845 controls. Clinical, para-clinical and lifestyle data were collected from each patient, as well as blood samples for genotyping the polymorphisms of four candidate genes for EH—MTHFR C677T (rs1801133), ACE I/D (*rs*4646994), AT1R A1166C (rs5186) and eNOS 4a/b—using PCR-based methods. **Results:** EH was associated with both genetic and environmental factors. Carriers of ACE DD and MTHFR TT genotypes presented an increased risk for EH (ACE DD: OR = 1.44, *p* = 0.0007; MTHFR TT: OR = 1.46, *p* = 0.0007). Lifestyle (smoking, physical activity) aspects were associated with EH. The risk of EH increased when both polymorphisms were associated with smoking (ACE DD: OR = 1.62, *p* = 0.0005; MTHFR TT: OR = 1.68, *p* = 0.0004). **Conclusions:** Our findings indicate that ACE I/D and MTHFR C677T may play a role in EH susceptibility, whereas polymorphisms AT1R A1166C and eNOS 4a/b do not appear to be associated. Furthermore, the interaction between genetic factors (ACE I/D, MTHFR C677T) and lifestyle factors such as smoking suggests an increased risk for developing essential hypertension.

## 1. Introduction

Arterial hypertension (AH) affects 1.28 billion of adults worldwide and 6.6 million adults in Romania [1]. A complex interplay between genetic and environmental factors contributes to both primary and secondary forms of hypertension. Approximately 95% of all cases of hypertension globally are identified as essential hypertension (EH), a complex multifactorial disease characterized by increased values for systolic blood pressure (SBP) and/or diastolic blood pressure (DBP) [2]. Given the fact that there is not a clearly identifiable cause for EH and patients with high blood pressure (HBP) are generally not investigated for an underlying etiology, only a small percentage are diagnosed with secondary hypertension. In secondary hypertension, HBP has an identifiable cause such as renal parenchymal disease, endocrine, renovascular or vascular disorders [3].

Genetic predisposition for EH is polygenic [4,5]. Certain mutations in methylenetetrahydrofolate reductase (MTHFR), angiotensin-converting enzyme (ACE), angiotensin II receptor type 1 (ATR1) and endothelial nitric oxide synthase (eNOS) genes may predispose to high blood pressure [6,7,8,9,10] and to an increased risk for EH, cardiovascular and cerebrovascular diseases.

MTHFR is involved in the metabolism of folate and homocysteine, which plays an essential role in DNA, RNA and protein methylation. Some mutations in *MTHFR* gene including MTHFR C677T were associated with hyperhomocysteinemia, which can predispose to cardiovascular diseases [11]. *MTHFR* gene is located on chromosome 1 (1p36.6) and the C677T polymorphism results from a mutation in the 4th exon, leading to the substitution of valine with alanine at codon 222 [12]. This mutation is associated with a reduced activity of the enzyme, with homozygous individuals showing increased homocysteine levels [11,12].

Angiotensin-1-converting enzyme (ACE) is a key component of the renin–angiotensin–aldosterone system (RAAS), which has a major role in regulating blood pressure and fluid–electrolyte balance [13]. Located on the long arm of the chromosome 17 (17q23), *ACE* gene presents multiple polymorphisms, the most studied being ACE I/D because of its strong association with cardiovascular diseases, including hypertension [14]. This insertion/deletion polymorphism that arises in the 16th intron influences the ACE activity in serum and defines three genotypes: insertion homozygous (II) (low activity), insertion/deletion heterozygous (ID) and deletion homozygous (DD) (high activity) types [15].

Another critical element of the RAAS is angiotensin II, a vasoconstricting peptide, that regulates blood pressure and sodium retention in the kidney through its binding to the angiotensin II type 1 receptor (AT1R). One of the polymorphisms in the *AT1R* gene, mapped on chromosome 3 (3q24), is A1166C, which consists of a single nucleotide change (A–C) at position 1166 in the 3′ untranslated region of the *AT1R* gene, influencing receptor expression [16]. Although A1166C has been linked with hypertension, the results obtained might vary depending on the population and the size of the samples [17,18].

In addition to the renin–angiotensin system, local autoregulatory mechanisms—mediated by endothelial cells and factors as endothelial nitric oxide (NO)—also contribute to the maintenance of blood pressure and blood flow [19]. The endothelial nitric oxide synthase 3 (*eNOS*) gene, located on chromosome 7 (7q35–7q36) is involved in the production of nitric oxide (NO), a biomolecule that induces relaxation of vascular tone, for which it is considered to have a vasoprotective effect. The *eNOS* gene is constitutively expressed in endothelial cells. The presence of eNOS variants attenuates NO production and might contribute to endothelial dysfunction. The 4a/b polymorphism is represented by a variable number of tandem repeats (VNTR) of a 27bp sequence in intron 4. The number of tandem repeats (four for variant “a” or five for variant “b”) may be an influencing factor for the gene’s activity [10]. Although studies on animal models suggest that endothelium-derived nitric oxide is involved in the regulation of blood pressure, the pattern of *eNOS* expression for patients with HBP has not been established.

The aim of this study was to assess the associations between EH and genetic variants MTHFR C677T, ACE I/D, AT1R A1166C and eNOS 4a/b in the adult Caucasian population of Romania.

## 2. Materials and Methods

### 2.1. Subjects

Power analyses were conducted using the PGA (Power for Genetic Association analyses) software to estimate the required sample size for this case–control study aiming to detect modest genetic effects. Simulations were performed under a recessive genetic model, assuming an odds ratio (OR) ranging from 1.5 to 1.75, a disease prevalence of 30%, a marker allele frequency of 0.2 and an effective degrees of freedom (EDF) of 1. A balanced study design was considered, with a 1:1 control-to-case ratio. Under these conditions, the study had 80% power to detect the specified genetic risk. The estimated number of cases required to achieve this power ranged from approximately 475 to 950, depending on the assumed effect size.

A total of 1690 subjects were included in this case–control study: 845 patients diagnosed with EH and 845 healthy normotensive individuals, aged between 35 and 65 years. The study enrollment was performed during 2015–2023 when participants were recruited from Bucharest and surrounding counties (Ilfov, Giurgiu, Dâmbovița, Prahova, Ialomița, Călarași) following visits to “Dr. Carol Davila” Central Military Emergency University Hospital and University Emergency Hospital Bucharest.

The inclusion and exclusion criteria were as follows. EH diagnosis was established based on a comprehensive evaluation, and the inclusion criteria for EH patients consisted of the following: elevated blood pressure values measured on three different occasions (SBP values were >140 mmHg and/or diastolic blood pressure values were >90 mmHg); administration of at least one anti-hypertensive agent (on average for 4 ± 2.5 years, for a period of 0–12 years); absence of any comorbidities (declared by participants) that are a criterion for secondary hypertension [3]. Patients with prehypertension (SBP between 120 and 140 mmHg and DBP between 80 and 90 mmHg) were not included in the study. In addition, no specific cause for EH was identified in patients by standard clinical methods. Therefore, the patients in this study were diagnosed with EH, without a cause being identified. At the time of recruitment, patients were not evaluated to determine the severity of hypertension.

Control subjects were included in the present study based on normal blood pressure values (less than 120/80 mmHg), the absence of any SH-related comorbidities, and no family history of cardiac diseases, including EH or AH, matching the patients in terms of gender, ethnicity and place of residence.

### 2.2. Genotyping

Genotyping was conducted at the Molecular Biology Laboratory of The National Institute of Research and Development for Food Bioresources.

Genomic DNA was extracted from 300 μL of peripheral blood collected on EDTA recipients using the Promega Wizard Genomic DNA Purification Kit (Promega Corporation, Madison, WI, USA). ACE I/D [20], eNOS 4a/b [21], AT1R A1166C [22] and MTHFR C677T [23] polymorphisms were genotyped using PCR-based methods as described in previous studies. PCR amplification was performed in a 20 μL reaction volume using Promega GoTaq^®^ Green Master Mix (Promega Corporation, Madison, WI, USA). Primer sequences, amplicons size and restriction enzymes used are presented in Table 1. After PCR, amplicons obtained for MTHFR C677T and AT1R A1166C polymorphisms were digested with restriction enzymes (*HinfI, TaqI* for MTHFR C677T; *DdeI* for AT1R); ACE I/D and eNOS amplicons did not require digestion. ACE I/D amplicons and restriction products for MTHFR C677T and AT1R A1166C were analyzed on polyacrylamide gel, while eNOS amplicons were analyzed on agarose gel for allele identification. Additionally, 10% of the samples were re-genotyped to ensure quality control.

### 2.3. Statistical Analysis

Chi-square test or Fisher’s exact test were used to compare the distribution of qualitative variables among groups. The associations between qualitative variables and EH were assessed by calculating odds ratio (OR) and 95% confidence interval (CI). Comparisons between groups of subjects were performed using the Mann–Whitney U test for non-normally distributed data. Stratified analysis was performed through χ^2^ test to evaluate the relationship between SNPs and qualitative variables including smoking and physical activity. We compared participants from the two groups (patients and controls) who carried a genotype of interest and were smokers (or sedentary) with those who did not carry this genotype and were non-smokers (or non-sedentary). The Bonferroni correction for 50 tests suggested that the *p* value for each test must be smaller than or equal to 0.001. The Hardy–Weinberg equilibrium (HWE) in the control group was evaluated by comparing observed and expected genotype frequencies using χ^2^ test. A *p*-value > 0.05 was considered consistent with HWE.

## 3. Results

### 3.1. Baseline Characteristics of the Subjects

For all the subjects included in this study, clinical data are presented in Table 2.

The mean age of subjects selected for this study (54.76 ± 6.96 years) was slightly higher in patients compared with controls, but not significant (54.97 ± 6.98 vs. 54.22 ± 6.94, *p* = 0.0046).

Smoking emerged as a risk factor for EH when an analysis between smokers and nonsmokers was performed. The frequency of smokers was higher in the EH group than in the control group (OR = 2.69, 95% CI = 2.2–3.29, *p* < 0.0001).

When subjects who had consumed or were currently consuming alcohol were compared with those who did not consume alcohol, no significant differences were identified between two groups.

Occasional or no physical activity emerged as a significant risk factor for the development of hypertension (OR = 2.45, 95% CI = 2.01–2.98, *p* < 0.0001), underlying the protective role of regular exercise in cardiovascular health.

Family history also appeared to play an important role in the pathology of EH. Subjects whose mothers had been diagnosed with hypertension exhibited a higher risk of developing EH (OR = 1.92, 95% CI = 1.33–2.78, *p* < 0.001).

### 3.2. Genotype Distribution

In the present study, we investigated the prevalence of variants among controls and EH patients. The results are presented in Table 3.

The genotype distribution within patient and control groups conformed to the Hardy–Weinberg equilibrium.

When comparing the EH patients and control groups, we found that EH was associated with ACE DD genotype (OR = 1.44, 95% CI: 1.17–1.77, *p* = 0.0007). The DD genotype was more common in the patient group than in the control group (33.6% vs. 26%), while the II genotype was more frequent in the control group than in the patient group (27.4% vs. 20.5%).

We have identified also a significant association between EH and MTHFR TT genotype (OR = 1.46, 95% CI: 1.17–1.82, *p* = 0.0007). Similarly, the frequency of the TT genotype was higher in the patient group compared to the control group (29.6% vs. 22.4%), whereas the CC genotype was more prevalent in the control group than in the patient group (29.7% vs. 23.8%).

No significant associations were identified for AT1R A1166C or eNOS (4a/b) polymorphisms when comparing the EH patients and control groups.

### 3.3. Gene–Environment Interactions

Since essential hypertension does not have an identifiable cause, we assessed potential gene–environment interactions. Therefore, we performed a stratified analysis based on the ACE DD and MTHFR TT genotypes and lifestyle factors for which we obtained significant associations with EH (smoking and physical activity).

Analysis of the interaction between physical activity and ACE DD and MTHFR TT genotypes did not reveal any significant associations. Conversely, significant interactions were observed between smoking and both risk genotypes, indicating that smoking may modulate genetic susceptibility to EH. We compared participants from the two groups (patients and controls) who carried the DD genotype of the *ACE* I/D polymorphism and were smokers with those who did not carry this genotype and were non-smokers. Similarly, for the *MTHFR* C677T polymorphism, participants with the TT genotype who were smokers were compared with those lacking this genotype and who were non-smokers. The results are presented in Table 4.

## 4. Discussion

Arterial hypertension is a heterogeneous disease with a complex etiology influenced by both genetic and environmental factors. Moreover, patients with hypertension have a higher rate of comorbidities [24] and a significant risk of cardiovascular complications, including death [25,26]. Most patients who present HBP without an identifiable cause are diagnosed with EH. The identification of an underlying cause would necessitate comprehensive investigations, a process that patients typically do not undergo. This under detection explains the low prevalence of secondary hypertension [2,3]. Therefore, in this study, we explored a variety of risk factors for EH, both genetic and non-genetic and the interplay between them.

We identified significant associations between EH and two genotypes: MTHFR TT and ACE DD.

MTHFR plays an essential role in the folate and homocysteine cycles. It catalyzes the conversion of 5,10-methylentetrahydrofolate into 5-methyltetrafolate, the main methyl-group donor for the conversion of homocysteine in methionine. Methionine is an essential sulfur-containing amino acid involved in cell differentiation, tissue development and organ function [11,12]. Altered methionine and methylation pathways have been associated with the development or progression of diseases such as diabetes and cancer [27]. Once the methionine cycle is initiated, metabolites such as S-adenosylmethionine are produced, which are crucial for methylation reactions of DNA, RNA, lipids, histones and other proteins [27,28].

The MTHFR C677T polymorphism is associated with reduced enzymatic activity, low levels of folate and high levels of homocysteine, which increased the risk of cardiovascular diseases. Meta-analysis studies have demonstrated a significant association between MTHFR C677T polymorphism and EH [29], especially among Caucasian and East Asian populations [30]. After evaluating 30 studies with 5207 cases and 5383 controls, a meta-analysis concluded that carriers of T allele or TT genotype presented an increased risk for EH [29], a conclusion that is consistent with our findings.

Furthermore, a recent cross-sectional and prospective cohort study performed in the UK Biobank analyzed the relationship between folate intake and hypertension [31]. From a total of 219,089 participants who did not have a hypertension diagnosis at enrolment, 17,670 participants developed hypertension after a median of 12.8 years. Participants with folate deficiency had a 42% higher risk of hypertension compared to those with normal folate levels, suggesting that folate deficiency might be a causal risk factor for hypertension.

Highly expressed in capillaries of the lung and the endothelium of the kidney, ACE plays a pivotal role in the regulation of blood pressure throughout RAS [32]. Its main function is to convert angiotensin I to angiotensin II, a potent vasoconstrictor [33].

The ACE I/D polymorphism consists of either an insertion (I) or deletion (D) of a 287 bp sequence within intron 16, accounting for variation in plasma ACE levels [34]. This polymorphism had been studied previously as a candidate marker for hypertension, but the results varied. One meta-analysis demonstrated that ACE I/D polymorphism is associated with an increased risk for EH, particularly with a higher frequency of D allele [35]. In addition, the DD genotype has been previously linked to greater susceptibility to hypertension [36], and several studies have further confirmed a correlation between the ACE I/D polymorphism and elevated risk of hypertension [37], myocardial infarction [38] and cardiovascular accident [39], but other studies have reported no such associations [40].

However, a limitation of the present study is that the plasma levels of markers affected, such as ACE, homocysteine and folate, were not measured, which would have allowed a more direct assessment of their functional implications.

Although previous studies have linked eNOS 4a/b [19] and AT1R A1166C [41] polymorphisms with increased risk of hypertension, our study did not find a statistically significant association with EH. This discrepancy may reflect characteristics of our cohort, including specific genetic background and environmental exposures, which could have limited the power to detect subtle effects.

Environmental factors also contribute significantly to the increased risk of essential hypertension. A sedentary lifestyle is one of the most associated risk factors for cardiovascular diseases. Current guidelines recommend at least 30 min of moderate-to-vigorous physical activity per day, five days a week [42]. Our study confirmed that the sedentary lifestyle or inconsistency of physical activity was associated with EH.

Unhealthy lifestyle factors such as smoking were associated with hypertension in previous studies [43,44], which aligns with the association observed in our study. Cigarette smoke can influence the folate cycle by reducing plasma folate levels [45]. It may also affect the methionine cycle, as compounds in cigarette smoke are associated with oxidative stress and deficiencies in essential cofactors (acid folic, vitamin B12). These alterations can elevate homocysteine levels, increasing cardiovascular risk [46].

Smoking has been associated with increased ACE activity, which leads to elevated angiotensin II levels. Acting primarily through AT1R, angiotensin II, a peptide that promotes vasoconstriction, activates pro-inflammatory and oxidative pathways, stimulating the production of reactive oxygen species (ROS). The release of ROS induces oxidative damage and lipid peroxidation, reducing the antioxidant capacity. All these changes contribute to endothelial dysfunction and therefore vasoconstriction and vascular impairment occur, amplifying the risk of cardiovascular disease [47,48].

The associations between EH and the MTHFR TT and ACE DD genotypes that we identified are consistent with previous findings reported in the literature, but the strength of these associations varies due to population-specific characteristics. Although routine genetic screening for hypertension is not currently recommended, identifying carriers of risk genotypes such as MTHFR TT or ACE DD could contribute to the development of personalized medicine strategies. These strategies might include targeted lifestyle interventions such as folate supplementation (in the case of MTHFR TT genotype carriers) and early monitoring of blood pressure or cardiovascular risk.

## 5. Conclusions

Our results suggest that both genetic factors (ACE DD and MTHFR TT genotypes) and lifestyle factors (smoking and physical activity) are associated with an increased risk of EH. Additionally, when the risk genotypes were correlated with smoking, the risk of EH increased, demonstrating an additive effect between genetic susceptibility and environmental exposure. No significant associations were identified for AT1R A1166C or eNOS (4a/b) polymorphisms.

These findings are supported by literature, and the observed variations may reflect population-specific characteristics, such as genetic diversity and stratification methods. They highlight the necessity for further studies investigating the interactions between genetic and environmental factors to better understand the etiology of essential hypertension and improve its diagnosis.

## Figures and Tables

**Table 1 biomedicines-13-02807-t001:** Primer sequences and amplicon size for the studied SNPs.

SNP	Primer Sequence	Amplicon Size
MTHFR C677T	5′ TGAAGGAGAAGGTGTCTGCGGGA 3′	198 bp (undigested)
AT1R	5′ AGGACGGTGCGGTGAGAGTG 3′	175 bp + 23 bp (after restriction)
	5′ GCACCATGTTTTGAGGTT 3′	546 bp (undigested)
	5′CGACTACTGCTTAGCATA 3′	435 bp + 111 bp (after restriction)
eNOS 4 a/b	5′ AGG CCC TAT GGT AGT GCC TTT 3′	393 bp (a allele)
	5′ TCT CTT AGT GCT GTG GTC AC 3′	420 bp (b allele)
ACE I/D	5′ CTGGAGACCACTCCCATCCTTTCT 3′	190 bp (without insertion)
	5′ GATGTGGCCATCACATTCG TCAGAT 3′	490 bp (with insertion)

SNP, single-nucleotide polymorphism.

**Table 2 biomedicines-13-02807-t002:** Characteristics of the subjects.

All Subjects	EH Patients (*n* = 845)	Controls (*n* = 845)	*p*	OR (95% CI)
Men/Women	462/383	433/412	0.16	
Age (mean ± SD) (year)	54.97 ± 6.98 (35–65)	54.22 ± 6.94 (35–65)	0.0046	
Height (mean ± SD) (m)	1.70 ± 0.03(1.58–1.87)	1.7 ± 0.04 (1.56–1.83)	<0.0001	
Weight (mean ± SD) (kg)	78.07 ± 8.89 (52–108)	69.28 ± 6.17 (54.0–88.0)	<0.0001	
BMI (mean ± SD) (kg/m^2^)	27.07 ± 3.01 (18.87–34.89)	23.86 ± 1.84 (19.15–29.41)	<0.0001	
Alcohol consumption status				
Non-consumers	576	609	0.34	
Current	29	79	<0.0001	0.34 (0.22–0.54)
Former	48	39	0.33	
Occasional	192	118	<0.0001	1.82 (1.41–2.33)
Smoking status				
Non-smokers	414	609	<0.0001	2.69 (2.2–3.29)
Current	141	79	<0.0001	0.51 (0.38–0.69)
Former	156	39	<0.0001	0.21 (0.15–0.31)
Occasional	134	118	0.31	
Physical activity status				
Occasionally active or sedentary	455	279	<0.0001	2.45 (2.01–2.98)
Moderately active	359	362	0.91	
Regularly active	31	204	<0.0001	0.12 (0.08–1.18)
Comorbidities				
T1DM	110/735	0		
T2DM	135/710	0		
Obesity	143/702	0		
MI	114/731	0		
CVA	147/698	0		
Family history				
Father with AH	119/726	87/758	0.02	
Mother with AH	86/759	47/798	<0.001	1.92 (1.33–2.78)

BMI, body mass index; SD, standard deviation, EH, essential hypertension; T1DM, type 1 diabetes mellitus; T2DM, type 2 diabetes mellitus; MI, myocardial infarction; CVA, cardiovascular accident; AH, arterial hypertension; OR, odds ratio; CI, confidence interval.

**Table 3 biomedicines-13-02807-t003:** Allele frequencies and genotype distribution of ACE I/D, MTHFR C677T, AT1R A1166C and eNOS 4a/b in EH patients and control groups.

SNP	Minor Allele	Alleles/Genotypes	HWE *p* Values in Controls	EH Patients (%) *n* = 845	Controls (%) *n* = 845	OR (95% CI)	*p*
ACE I/D (rs4646994)	I	I	0.058	734 (43.4)	856 (50.6)	0.75 (0.65–0.86)	<0.001
		D		956 (56.6)	834 (49.4)	1.34 (1.17–1.53)	
		I/I		173 (20.5)	231 (27.4)	0.68 (0.55–0.85)	0.001 *
		I/D		388 (45.9)	394 (46.6)	0.97 (0.80–1.18)	0.76
		D/D		284 (33.6)	220 (26.0)	1.44 (1.17–1.77)	0.0007 *
MTHFR C677T (rs1801133)	T	C	0.29	796 (47.1)	907 (53.7)	0.77 (0.67–0.88)	0.0001
		T		894 (52.9)	783 (46.3)	1.30 (1.14–1.49)	
		C/C		201 (23.8)	251 (29.7)	0.74 (0.60–0.92)	0.044
		C/T		394 (46.6)	405 (47.9)	0.95 (0.78–1.15)	0.60
		T/T		250 (29.6)	189 (22.4)	1.46 (1.17–1.82)	0.0007 *
AT1R A1166C (rs5186)	C	A	0.31	1253 (74.1)	1215 (71.9)	1.12 (0.96–1.31)	0.14
		C		437 (25.9)	475 (28.1)	0.89 (0.77–1.04)	
		A/A		454 (53.7)	431 (51.0)	1.12 (0.92–1.35)	0.26
		A/C		345 (40.8)	353 (41.8)	0.96 (0.79–1.17)	0.69
		C/C		46 (5.5)	61 (7.2)	0.74 (0.5–1.1)	0.14
eNOS (4a/b)	a	A	0.08	392 (23.2)	382 (22.6)	1.03 (0.88–1.21)	0.68
		B		1298 (76.8)	1308 (77.4)	0.97 (0.82–1.14)	
		Aa		48 (5.7)	34 (4.0)	1.44 (0.92–2.26)	0.11
		Ba		296 (35.0)	314 (37.2)	0.91 (0.75–1.11)	0.36
		Bb		501 (59.3)	497 (58.8)	1.02 (0.84–1.24)	0.84

SNP, single-nucleotide polymorphism; HWE, Hardy–Weinberg equilibrium; EH, essential hypertension; OR, odds ratio; CI, confidence interval; * *p* ≤ 0.001.

**Table 4 biomedicines-13-02807-t004:** Association analysis of ACE I/D and MTHFR C677T genotypes with smoking status.

SNP	Genotype	EH Patients	Controls	OR (95 CI%)	*p*
ACE I/D	DD	152	101	1.62 (1.23–2.12)	0.0005 *
	II + ID	693	744		
MTHFR C677T	TT	138	88	1.68 (1.25–2.26)	0.0004 *
	CC + CT	707	757		

SNP, single-nucleotide polymorphism; EH, essential hypertension; OR, odds ratio; CI, confidence interval, * *p* ≤ 0.001.

## Data Availability

The data presented in this study are available on request from the corresponding author. The data are not publicly available due to privacy and ethical issues.

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
