# Peer review of "Analysis of Genetic Variants MTHFR C677T, ACE I/D, AT1R A1166C and eNOS 4a/b in the Context of Essential Hypertension Susceptibility"

_biomedicines, 2025, doi:10.3390/biomedicines13112807_

Round 1

Reviewer 1 Report

Comments and Suggestions for Authors

In the present study authors analyzed selected polymorphisms of genes involved in the regulation of cardiovascular system in adult Romanian patients with essential hypertension in comparison to healthy control subjects. The results show that ACE I/D and MTHFR C677T polymorphims are associated with hypertension whereas At1R and eNOS polymorphims are not.

The topic is of interest, however, there are also some important concerns to be addressed.

  1. Line 58, MTHFR is directly involved in the metabolism of folate and due to metabolism of folate also in homocysteine metabolism. Therefore, the text should be modified to "folate and homocysteine" rather than: "homocysteine and folate".
  2. Line 74, AT1R is not the vasoconstricting peptide but the receptor for angiotensin II, the latter being the vasoconstricting peptide.
  3. Section 2.1, was the required number of patients calculated before the study?
  4. LIne 109, mean age and SD/SEM should be presented with the same precision (either 1 or 0.1 years); 0.01 years is about 3.5 days which makes no sense.
  5. Line 110, how were the causes of secondary hypertension excluded? Were only specialistic investigations performed?
  6. Section 2.1, according to the inclusion criteria for both groups patients with prehypertension (SBP between 120 and 140 mmHg and DBP between 80 and 90 mmHg) were not included, am I right?
  7. Line 123, the method of genotyping should be described as the main method of this study.
  8. Line 160, how was consistency with Hardy-Weinberg equilibrium verified?
  9. Section 3.3, the method used to analyze the interaction between the genotype and smoking/physical activity should be described in the Methods.
  10. Was homocysteine level measured in these patients?
  11. Line 241, it should be clarified how increased ACE activity could increase ROS.

Author Response

All the changes are marked in yellow in the revised manuscript. Please see the attachment.

Comments 1: Line 58, MTHFR is directly involved in the metabolism of folate and due to metabolism of folate also in homocysteine metabolism. Therefore, the text should be modified to "folate and homocysteine" rather than: "homocysteine and folate".

Response 1: Thank you for pointing this out. We agree with this comment. Therefore, we have made the change – page 2, and line 60.]

Comments 2: [Line 74, AT1R is not the vasoconstricting peptide but the receptor for angiotensin II, the latter being the vasoconstricting peptide.]

Response 2: Agree. We have, accordingly, modified this: Another critical element of the RAAS is angiotensin II, a vasoconstricting peptide, that regulates blood pressure and sodium retention in the kidney through its binding to the angiotensin II type 1 receptor (AT1R). page 2, lines 76-78

Comment 3: Section 2.1, was the required number of patients calculated before the study?

Response 3: Yes, it was. We included the paragraph in the manuscript - Materials and Methods, lines 101 - 110: 

Power analyses were conducted using the PGA (Power for Genetic Association analyses) software to estimate the required sample size for this case-control study aiming to detect modest genetic effects. Simulations were performed under a recessive genetic model, assuming an odds ratio (OR) ranging from 1.5 to 1.75, a disease prevalence of 30%, a marker allele frequency of 0.2, and an effective degrees of freedom (EDF) of 1. A balanced study design was considered, with a 1:1 control-to-case ratio. Under these conditions, the study had 80% power to detect the specified genetic risk. The estimated number of cases required to achieve this power ranged from approximately 475 to 950, depending on the assumed effect size.

Comment 4: LIne 109, mean age and SD/SEM should be presented with the same precision (either 1 or 0.1 years); 0.01 years is about 3.5 days which makes no sense.

Response 4: Agree. We made the changes in line 121 and Table 2.

Comment 5: Line 110, how were the causes of secondary hypertension excluded? Were only specialistic investigations performed?

Response 5: Based on the information participants declared - line 122

Comment 6: Section 2.1, according to the inclusion criteria for both groups patients with prehypertension (SBP between 120 and 140 mmHg and DBP between 80 and 90 mmHg) were not included, am I right?

Response 6: Yes, we included this information in the manuscript - lines 123 - 124

Comment 7: Line 123, the method of genotyping should be described as the main method of this study.

Response 7: Agree, thank you for highlighting this. We included the following paragraph and a table with primer sequences and amplicon size: PCR amplification was performed in a 20 μl reaction volume using Promega GoTaq® Green Master Mix. Primer sequences, amplicons size and restriction enzymes used are presented in Table 1. After PCR, amplicons obtained for MTHFR C677T and AT1R A1166C polymorphisms were digested with restriction enzymes (HinfI, TaqI for MTHFR C677T; DdeI for AT1R); ACE I/D and eNOS amplicons did not require digestion. ACE I/D amplicons and restriction products for MTHFR C677T and AT1R A1166C were analyzed on polyacrylamide gel, while eNOS amplicons on agarose gel for allele identification. Additionally, 10% of the samples were re-genotyped to ensure quality control. lines 139 - 147

Comment 8: Line 160, how was consistency with Hardy-Weinberg equilibrium verified?

Response 8: 

The Hardy-Weinberg equilibrium (HWE) in the control group was evaluated by comparing observed and expected genotype frequencies using χ² test. A p-value > 0.05 was considered consistent with HWE. lines 161 - 163

Comment 9: Section 3.3, the method used to analyze the interaction between the genotype and smoking/physical activity should be described in the Methods.

Response 9: Agree - Stratified analysis was performed through χ² test to evaluate relationship between SNPs and qualitative variables including smoking and physical activity. We compared participants from the two groups (patients and controls) who carried a genotype of interest and were smokers (or sedentary) with those who did not carry this genotype and were non-smokers (or non-sedentary). lines 157-161

Comment 10: Was homocysteine level measured in these patients?

Response 10: No, we did not perform any measurements from enzymatic activity, therefore we included this aspect as a limitation of this study.

Comment 11: Line 241, it should be clarified how increased ACE activity could increase ROS.

Response 11: Agree. Please see the extended paragraph, lines 295-301: 

Smoking has been associated with increased ACE activity, which leads to elevated angiotensin II levels. Acting primarily through AT1R, angiotensin II, a peptide that promotes vasoconstriction, activates pro-inflammatory and oxidative pathways, stimulating the production of reactive oxygen species (ROS). The release of ROS induces oxidative damage and lipid peroxidation, reducing the antioxidant capacity. All these changes contribute to endothelial dysfunction and therefore vasoconstriction and vascular impairment occur, amplifying the risk of cardiovascular disease [48, 49].

Reviewer 2 Report

Comments and Suggestions for Authors

This study entitled “Analysis of Genetic Variants MTHFR C677T, ACE I/D, AT1R 2 A1166C, eNOS 4a/b in the Context of Essential Hypertension Susceptibility” by Remus Nica et al. investigates genetic and environmental risk factors contributing to essential hypertension (EH). The authors identified significant associations between EH and two genetic polymorphisms: MTHFR C677T (TT genotype) and ACE I/D (DD genotype). Additionally, lifestyle factors such as physical inactivity and smoking were identified as major environmental contributors to EH, with smoking influencing folate metabolism, oxidative stress, and endothelial function.

  • Authors did not provide experimental evidence such as enzyme assays to support functional implications, at least this should be included as a limitation in Discussion section.
  • Relate findings to global data and discuss potential implications for personalized medicine or genetic screening for hypertension risk.

Author Response

All changes are marked in green in the revised manuscript.

Comment 1: Authors did not provide experimental evidence such as enzyme assays to support functional implications, at least this should be included as a limitation.

Response: Agree. We included this as a limitation, lines 278 - 280: However, a limitation of the present study is that the plasma levels of markers affected, such as ACE, homocysteine and folate, were not measured, which would have allowed a more direct assessment of their functional implications.

Comment 2: Relate findings to global data and discuss potential implications for personalized medicine or genetic screening for hypertension risk.

Response 2: Agree. Thank you for pointing this out. We added two paragraphs, lines 256-265: 

After evaluating 30 studies with 5207 cases and 5383 controls, a meta-analysis concluded that carriers of T allele or TT genotype presented an increased risk for EH [29], a conclusion that is consistent with our findings.

Furthermore, a recent cross-sectional and prospective cohort study performed in the UK Biobank analyzed the relationship between folate intake and hypertension [31]. From a total of 219 089 participants who did not have a hypertension diagnosis at enrolment, 17 670 participants developed hypertension after a median of 12.8 years. Participants with folate deficiency had a 42% higher risk of hypertension compared to those with normal folate levels, suggesting that folate deficiency might be a causal risk factor for hypertension. 

and lines 304 - 311: The associations between EH and the MTHFR TT and ACE DD genotypes we identified are consistent with previous findings reported in the literature, but the strength of these associations varies due to population-specific characteristics. Although routine genetic screening for hypertension is not currently recommended, identifying carriers of risk genotypes such as MTHFR TT or ACE DD could contribute to the development of personalized medicine strategies. These strategies might include targeted lifestyle interventions such as folate supplementation (in the case of MTHFR TT genotype carriers) and early monitoring of blood pressure or cardiovascular risk.

Reviewer 3 Report

Comments and Suggestions for Authors

This manuscript investigated the associations between essential hypertension (EH) and genetic polymorphisms (MTHFR C677T, ACE I/D, AT1R A1166C, eNOS 4a/b), as well as their gene-environment interactions, in an adult Caucasian population from Romania. This study contributes valuable population-specific evidence to the understanding of EH etiology. However, there are several issues:

1, They found that ACE I/D and MTHFR C677T may play a role in EH susceptibility, which consistent with the results of many previous studies.

2, Please show the exact p value for those p > 0.001.

3, Please add the detailed information of arterial hypertension. Is there any subgroup or the classification of different severity?

4, Height and weight might be another important factors.

4, It would be better to include the follow-up information, to confirm the relationship between these genetic polymorphisms and hypertension.

Author Response

All changes made are highlighted in blue. Please see attachment.

Comment 1: They found that ACE I/D and MTHFR C677T may play a role in EH
susceptibility, which consistent with the results of many previous studies.

Response 1: Yes

Comment 2: Please show the exact p value for those p > 0.001.
Response 2:
Please see Table 2 for revised values.

Comment 3: Please add the detailed information of arterial hypertension. Is there any subgroup or the classification of different severity?

Response 3: No, and thank you for pointing this out. We included this information, lines 126 - 127: At the time of recruitment, patients were not evaluated to determine the severity of hypertension.

Comment 4: Height and weight might be another important factors.

Response 4: Please see Table 2 for these two variables added.

Comment 5: It would be better to include the follow-up information, to confirm
the relationship between these genetic polymorphisms and hypertension."

Response 5: An excelent observation, but unfortunately this was not longitudinal study. 

Round 2

Reviewer 1 Report

Comments and Suggestions for Authors

The manuscript has been revised according to the reviewers' comments. All concerns raised by the reviewers have been adequately addressed by the authors. 

Author Response

Comment: The manuscript has been revised according to the reviewers' comments. All concerns raised by the reviewers have been adequately addressed by the authors. 

Response: We would like to thank you for your valuable comments and suggestions, which have helped improve the quality of our manuscript.

Reviewer 2 Report

Comments and Suggestions for Authors

Authors have addressed the changes pointed by this reviewer

Author Response

Comment: Authors have addressed the changes pointed by this reviewer.

Response: We would like to thank you for your valuable comments and suggestions, which have helped improve the quality of our manuscript.

Reviewer 3 Report

Comments and Suggestions for Authors

1, Table 2, the height between two groups are quite similar but the p value is <0.0001, please check.

2, Family history also appeared to play an important role in the pathology of EH. Subjects whose mothers had been diagnosed with hypertension exhibited a higher risk of developing EH (OR=1.92, 95%CI = 0.25–0.60, p<0.001).

The OR here conflicts with the 95% CI. Please clarify.

3, The mean age of subjects selected for this study (54.76±6.96 years) was slightly higher 177 in patients compared with controls (54.97±6.98 vs. 54.22±6.94, p=0.0046).

This is inconsistent with Table 2.

Author Response

Comment 1: Table 2, the height between two groups are quite similar but the p value is <0.0001, please check.

Response 1: Thank you for your observation. The two groups are not normally distributed, so we used Mann Whitney U test. We verified the result – the p value is 0.000033 (<0.0001).

Comment 2: Family history also appeared to play an important role in the pathology of EH. Subjects whose mothers had been diagnosed with hypertension exhibited a higher risk of developing EH (OR=1.92, 95%CI = 0.25–0.60, p<0.001).

The OR here conflicts with the 95% CI. Please clarify.

Response 2: Thank you for your observation, we agree. The correct values for 95%CI were included in Table 2, we updated the values in the text also. Please see lines 189 – 190.

Comment 3: The mean age of subjects selected for this study (54.76±6.96 years) was slightly higher 177 in patients compared with controls (54.97±6.98 vs. 54.22±6.94, p=0.0046).

This is inconsistent with Table 2.

Response 3: Agree, thank you for bringing this to our attention. The correct value is the one included in text, so we corrected the value corresponding to age in Table 2. Please see the line for age, Table 2. We also specified that this difference is not significant in the main text. Please see line 177: „The mean age of subjects selected for this study (54.76±6.96 years) was slightly higher in patients compared with controls, but not significant (54.97±6.98 vs. 54.22±6.94, p=0.0046).”

Round 3

Reviewer 3 Report

Comments and Suggestions for Authors

I recommend that this manuscript undergo further review by a statistician with expertise in relevant analytical methods. The authors have not addressed the concerns I raised in my two previous review reports. Key issues remain unresolved, and a specialized statistical assessment is necessary to verify the validity of the study’s methodology and results.